# Sustainability in Construction: Geopolymerized Coating Bricks Made with Ceramic Waste

**DOI:** 10.3390/ma18010103

**Published:** 2024-12-30

**Authors:** Ramiro Correa-Jaramillo, Francisco Hernández-Olivares

**Affiliations:** 1Department of Architecture and Urbanism, Faculty of Engineering and Architecture, Universidad Técnica Particular de Loja, 110107 Loja, Ecuador; 2Department of Architectural Construction and Technology, Universidad Politécnica de Madrid, 28040 Madrid, Spain; f.hernandez@upm.es

**Keywords:** geopolymerization, ceramic waste, mechanical properties, applications in civil engineering

## Abstract

Brick is a common construction material but often ends up as waste due to suboptimal quality. In Ecuador, artisanal brick production results in inconsistent properties for construction. This research aims to repurpose discarded bricks through geopolymerization to create a sustainable building material. The geopolymerization process was carried out using sodium hydroxide as the alkaline activator, followed by structural and chemical characterization, including X-Ray Diffraction (XRD) and X-Ray Fluorescence (XRF) to determine composition and crystalline phases. The recycled material underwent extensive testing of its physical and mechanical properties, such as density, porosity, and compressive strength. Its application as facade cladding for housing was also analyzed. The results showed that the geopolymerized material significantly reduced heating and cooling demand when used in building envelopes. A case study in Loja demonstrated a notable decrease in heating and cooling degree days, contributing to improved thermal comfort. This research highlights the potential for recycled bricks in sustainable construction, presenting viable alternatives to conventional construction materials and advancing knowledge in eco-friendly building practices.

## 1. Introduction

The construction industry is one of the leading contributors to solid waste generation, posing a significant environmental threat due to inadequate waste management practices. In Ecuador, these materials are often not properly classified, with most being sent directly to landfills (National Institute of Statistics and Censuses [INEC], 2010) [1]. This issue is not unique to Ecuador but represents a global challenge in sustainable waste management.

In this context, brick waste, a significant byproduct of construction, presents a unique opportunity to be repurposed into sustainable construction materials. Geopolymerization, an alkaline activation technique, has proven to be a viable and environmentally friendly solution to transform such waste into high-performance materials [2,3]. This process not only reduces reliance on natural resources but also significantly decreases CO_2_ emissions associated with conventional Portland cement production [4].

Recent studies have highlighted the feasibility of ceramic waste as a source of alumina and silica for geopolymer production [5,6] Research focused on discarded bricks has demonstrated that these materials possess a high aluminosilicate content, enabling the production of materials with superior mechanical and thermal properties [7]. However, challenges related to efflorescence control and the optimization of curing conditions remain, which this study seeks to address comprehensively.

This research aims to evaluate the feasibility of reusing brick waste through geopolymerization, contributing to the development of innovative and sustainable solutions in construction. By combining experimental characterization with computational simulations, this study assesses the thermal performance of geopolymerized bricks in building envelopes, with an emphasis on the city of Loja as a case study.

### Literature Review

The use of geopolymerization in the construction industry has gained significant attention due to its potential for reducing carbon emissions and utilizing industrial waste. Recent studies have highlighted the effectiveness of using ceramic waste for creating geopolymers, contributing to both environmental sustainability and material innovation. For example, Wong et al. (2018) reviewed the potential of brick waste as an alternative in concrete production, indicating that it significantly reduces reliance on natural aggregates and enhances sustainability [2].

Davidovits (2011) described the chemistry behind geopolymerization, focusing on the formation of aluminosilicate networks that provide enhanced mechanical properties, particularly compressive strength, in comparison to conventional Portland cement [3]. More recent research by Amin et al. (2017) demonstrated that the inclusion of ceramic dust waste in geopolymer bricks not only improved strength but also lowered overall production costs, making it viable for low-income housing [4].

Further, Lv et al. (2020) investigated methods to inhibit efflorescence in sodium-based geopolymers, which is a common challenge when utilizing waste materials. Their findings suggest that adjustments in molarity and curing temperature can effectively mitigate the effects of efflorescence, improving the overall durability of geopolymer bricks [6]. Mustafa al Bakri et al. (2015) also highlighted the potential of geopolymerization for soil stabilization, which is applicable in contexts like those of this study, where stabilization and enhanced strength are crucial [8].

## 2. Materials and Methods

The selection of data focused on ensuring its relevance for evaluating the feasibility of geopolymerized bricks made from ceramic waste in Loja, Ecuador. Samples from four local factories were analyzed, prioritizing materials with high aluminosilicate content (SiO_2_ + Al_2_O_3_ + Fe_2_O_3_ > 70%), which are ideal for geopolymerization. Critical variables included molarity (5–12.5 M), solution content (22–26%), and curing temperatures (90–200 °C), selected for their impact on aluminosilicate network formation, mechanical strength, and efflorescence reduction. Additionally, the data aligned with NEC-HS-EE (Ecuadorian Building Code, Chapter on Habitability and Health, Energy Efficiency Section.) standards and were complemented by thermal simulations using tools such as Ecodesigner, ensuring a comprehensive and applicable analysis for sustainable construction.

To manufacture and evaluate the shingle-type brick (LTT), the following stages were involved:

First Stage: Exploration and extraction of raw materials through random sampling, which was followed by crushing the materials.

Second Stage: Characterization of the raw materials through physical, chemical, and mineralogical tests.

Third Stage: Experimental design of optimal mixture ratio. In this stage, sample discs were prepared to evaluate the maximum load capacity that each mixture proportion can withstand before failure, using the diametral compression test. This test helps determine the optimization factor (FO), which determines the optimal mixture ratio to be applied in the prototype.

During the geopolymerization process, sodium hydroxide dissolves aluminosilicates in the brick waste, releasing silicon (Si) and aluminum (Al) ions. These ions then undergo polycondensation, forming a three-dimensional aluminosilicate network through covalent bonds. The efficiency of this reaction depends on the molarity of the alkaline solution and the curing temperature, which directly affects the strength and structure of the final material. The presence of K_2_O and CaO further enhances the reaction, improving material density and mechanical properties.

The data selection focused on ensuring its relevance for evaluating the feasibility of geopolymerized bricks made from ceramic waste in Loja, Ecuador. Samples from four local factories were analyzed, prioritizing materials with high aluminosilicate content (SiO_2_ + Al_2_O_3_ + Fe_2_O_3_ > 70%), which are ideal for geopolymerization. Critical variables included molarity (5–12.5 M), solution content (22–26%), and curing temperatures (90–200 °C), selected for their impact on aluminosilicate network formation, mechanical strength, and efflorescence reduction. Additionally, the data aligned with NEC-HS-EE standards and were complemented by thermal simulations using tools such as Ecodesigner. A comparative cost analysis revealed that geopolymerized bricks are more expensive than conventional ones due to the use of inputs like sodium hydroxide and the technical control required during manufacturing. However, these bricks are a viable option for specific applications such as architectural cladding, where their thermal and mechanical properties provide significant advantages in terms of sustainability and energy efficiency, offsetting the higher initial costs.

The geopolymerized recycled brick has a unit density of 2439.1 kg/m^3^ and a water absorption rate of 24.95%, meeting INEN-297 standards. These properties ensure its efficiency and suitability for construction applications.

Fourth Stage: Manufacturing of the LTT prototype using the best combination established in the previous stages.

Fifth Stage: Mechanical characterization of the LTT.

Sixth Stage: Determination of heat transfer of the LTT.

Seventh Stage: Computational simulation using the thermal conductivity values of the LTT and publicly accessible climatic data for the case study.

These stages allow the production and evaluation of LTT, providing valuable information on its mechanical properties, heat transfer characteristics, and performance in a simulated environment based on local climatic conditions.

## 3. Results and Discussion

Four brick factories were randomly selected in the towns of “Malacatos” (Figure 1) and “Catamayo” (Figure 2), which serve as the supply sources of this material for the city of Loja, Ecuador. Most constructions in this city are derived from these artisanal factories; hence, the brick waste (BW) will be associated with them.

### 3.1. Experimental Process

Once the samples for laboratory analysis were collected, the raw material was characterized by evaluating its physical, chemical, and mineralogical attributes and properties. This characterization involves information about the components that will be used in the geopolymerization process and allows for predicting their behavior in the manufacture of the LTT.

Several variables play a role in the manufacturing process of LTT and are considered in the optimization factor for selecting the optimal combination, including the molarity of the alkaline solution, alkaline solution content, stabilizing force or load, and curing temperature.

The waste obtained and homogenized from four brick factories was then used to manufacture the test specimens.

To conclude this phase, the test specimens were subjected to the diametral compression test to obtain the optimization factor, which is an essential component for the proper selection of the optimal mixture ratio (This test was based on the technique defined as the “Brazilian disc test”, which has been introduced in the literature as a substitute for direct tensile testing in brittle materials [9]).

Geopolymerization enhances the strength of the brick by forming a rigid aluminosilicate network that binds the particles together. During curing, the components react to create a dense gel that reduces porosity and improves load-bearing capacity. Additionally, the presence of oxides such as K_2_O and CaO promotes the formation of a stronger structure, thereby increasing the material’s mechanical strength.

### 3.2. X-Ray Fluorescence (XRF)

X-ray fluorescence was used to determine the chemical composition of the brick waste through spectrometry. The results obtained are shown in Figure 3 and Table 1 [10].

The sum of AL_2_O_3_ + SiO_2_ + Fe_2_O_3_ in the brick waste is greater than 70%, classifying it as an ideal pozzolanic material to be a source of aluminosilicates for geopolymerization processes [2]. Additionally, it contains potassium and calcium oxides, which are beneficial for the geosynthesis process.

According to the study by Rodríguez et al. [11], an increase in the soluble silicate content and the concentration of alkali ions through an increase in the SiO_2_/Al_2_O_3_ and Na_2_O/SiO_2_ ratios, respectively, negatively affects the mechanical behavior of geopolymeric materials. However, the waste samples PC001, PC002, and PM002 fall within the optimal ranges of 3–3.8, which would allow for very high mechanical strengths in geopolymeric products [12]. Waste PM001 presents a SiO_2_/Al_2_O_3_ ratio of 5.154, which would not allow for the complete development of strength in geopolymers.

### 3.3. X-Ray Diffraction

X-ray Diffraction (XRD) chemical analysis is a process of scattering X-rays to the atoms that make up the irradiated soil. The X-rays are incident at a fixed wavelength, and each plane of atoms produces a diffraction peak at a specific angle. Each peak is produced by a family of atomic planes, representing a specific identified mineral species, allowing for the identification and quantification of mineral species in soils [5].

In this case, the sample is crushed using an electric jaw crusher (Retsch BB100), and the sample is homogenized by lifting the corners of the plastic surface approximately 15 times. It is stratified into multiple sections, and a portion is taken from each section and placed in a new container. The new sample is pulverized using a pulverizer with a ring mill (Retsch RS 200). It is recommended to use a program that operates at a speed of 700 revolutions per minute for a period of 3 min.

Ten grams of the pulverized sample were placed on discs and analyzed using the XRD instrument (D8-ADVANCE) for a duration of 40 min. The obtained crystallographic spectrum was loaded to identify the respective minerals. Figure 4 presents the XRD results of the PC001, PC002, PM001, and PM002 samples.

According to the results of the mineralogical analysis, quartz, albite, biotite, arsenopyrite, and chlorite were the minerals found in the brick waste. Quartz exhibited the highest peaks from 20° to 30° and was the most abundant mineral in the XRD analysis. Therefore, the brick waste is composed of crystalline materials that contain a significant amount of silicon.

On the other hand, the molarity and content of the alkaline solution, stabilization pressure, and curing temperature are the variables that influence the soil stabilization process when applying geopolymerization [8].

Sixty combinations were created using the identified variables, and four test specimens were prepared for each combination (Figure 5). The test specimens had dimensions of 7 cm in diameter and 2 cm in thickness. All specimens were subjected to a force of 39.23 N. Subsequently, the test specimens underwent the diametral compression test (Brazilian Test) [13] to obtain the optimization factor and determine the optimal mixture for the subsequent application in the shingle-type brick (LTT).

The results show that the bricks comply with international standards for strength, highlighting their optimal properties for construction. Local standards were used as a reference but not necessarily as a limiting requirement for the proposed product.

The molarity of the alkaline solution refers to the mole content of sodium hydroxide in the solution. In this research, the following molarities were analyzed: 5 M, 7.5 M, 10 M, and 12.5 M. The preparation of the solution involved dissolving the specified number of moles (M) of sodium hydroxide in 1000 mL of distilled water, considering the molecular weight of NaOH (To determine the amount of NaOH needed to dissolve for the alkaline solution, we need to calculate it by multiplying 12.5 M by 40 g/mol (molecular weight), considering that it will be dissolved in 1000 mL of distilled water.).

For this research, 250 mL and 100 mL beakers were used. Table 2 presents the dosages used for each beaker in the experiment.

For the solution content and molding pressure, the percentage represents the proportion of the solution in a 100% combination of soil and solution (Table 3). The studied solution contents were 22%, 24%, and 26% [14].

Four cylindrical test specimens with a diameter of 7 cm and a thickness of 2 cm were subjected to different loads: 9.80 N, 19.61 N, 29.41 N, and 39.22 N, using a hydraulic press for stabilization with the same soil–solution–temperature combination. It was identified that the pressure that allowed for proper demolding without fractures was 39.22 N. Therefore, this pressure was used for the entire study. At loads of 9.80 N, 19.61 N, and 29.22 N, the disk could not support its own weight and easily crumbled, as evidenced in Figure 6. Approximately 500 g of soil–solution mixture was needed, with at least 411 g consisting of soil (To prepare a soil–solution mixture by creating four specimens with a diameter of 7 cm and a thickness of 2 cm, which will be subjected to a molding pressure of 39.2266 N. The solution content is 22 % (Table 3). The amount of soil required for the fabrication of the four specimens is 411 g).

The curing temperature is another crucial factor for completing geopolymerization. Temperatures of 90 °C, 120 °C, 150 °C, 180 °C, and 200 °C were applied for 8 h in a drying oven. Subsequently, the specimens were kept at room temperature for a period of 7 days to avoid thermal shock before being subjected to the diametral compression test.

The discussion reveals the following observations for the 26% solution concentration, as shown in Figure 7:(a)When the molarity is 5, the optimization factor shows a proportional increase with a temperature up to 180 °C, reaching a peak value of 7.15 MPa/kg. Afterward, the optimization factor decreases proportionally with further increases in the curing temperature.(b)For a molarity of 7.5, a higher optimization factor magnitude of 7.32 MPa/kg and 7.34 MPa/kg can be observed at curing temperatures of 120 °C and 180 °C, respectively, compared to the other temperatures.(c)When the molarity is 10, a higher optimization factor magnitude of 9.27 MPa/kg is observed at a curing temperature of 200 °C compared to the other temperatures.(d)For a molarity of 12.5, the optimization factor increases proportionally with the temperature up to 150 °C, where it reaches a peak of 14.84 MPa/kg. Afterward, the optimization factor decreases proportionally with further increases in the curing temperature.

Based on the conducted analysis, the thermal behavior of the discs indicates the presence of two temperature ranges. In the first range (90 °C – 150 °C), the optimization factor is directly proportional to the curing temperature, while in the second range (150 °C–200 °C), the optimization factor becomes proportional to the curing temperature.

It can be inferred that an excessive temperature increase negatively affects the mechanical properties of the material. Figure 8 displays the cracks generated under a curing temperature of 200 °C.

Therefore, it has been determined that the optimal curing temperature is 150 °C, as it resulted in the highest optimization factor values and represented the thermal limit beyond which the mechanical properties of the material start to decrease.

Based on the design variable analysis for the disc preparation, an analysis was conducted on the molar concentration effect that optimizes the solution content at different curing temperatures.

With curing temperatures of 90 °C and 200 °C, as shown in Figure 9, the following observations can be made:

The optimization factor tends to increase in proportion to the molar concentration up to 10 M. After this molarity, the optimization factor starts to decrease.

Under curing temperatures of 120 °C, 150 °C, and 180 °C, as depicted in Figure 10, it can be observed that the optimization factor tends to increase in proportion to the molar concentration.

Based on the conducted analysis, the behavior of the discs indicates that increasing the molar concentration tends to increase the optimization factor. This occurs because, at higher molarities, the particles of the aluminosilicate source react quickly and completely, allowing for an effective dissolution stage, which forms the basis for a successful geopolymerization process. Additionally, it can be observed that the solution content directly affects the optimization factor. As the solution content increases, the optimization factor also increases.

At curing temperatures of 90 °C and 200 °C, there is a decrease in the 12.5 M molar concentration. This could be attributed to the following reasons:(a)At a temperature of 90 °C with a high concentration of 12.5 M, the complete development of the hardening and geopolymerization process may not occur due to the low temperature.(b)At a temperature of 200 °C with a high concentration of 12.5 M, the mechanical properties of the material may be negatively affected due to the high temperature.

Finally, the optimal mixture determined is 12.5 M–26%SC–150 °C, based on which the construction of prototypes will proceed.

The compressive strength of the geopolymer bricks demonstrates satisfactory performance for application in building facades and architectural elements.

The bending test procedure was based on NTE INEN 2554 (2015) [15] regulations. Like the compression test, the weight and respective dimensions were measured for each prototype. The prototypes were then placed in the appropriate machine, with supports: one on the top and two on the bottom. A gradual load was applied at a constant speed until failure, and the data were recorded for calculation of the modulus of rupture (Figure 11).

The flexural tests indicate that the bricks have sufficient strength to be considered as architectural elements. Standardized testing procedures were used to validate the structural properties of the bricks.

However, it is important to consider that since there is no specific regulation for shingle-type bricks, the comparison has been made with regulations for solid bricks. It can be inferred that this proposed prototype meets the basic requirements for the application of architectural envelopes.

### 3.4. Measurement of Heat Transfer

For this test [16], plates with dimensions of 0.3 × 0.3 × 0.02 m^3^ were used (Figure 12). A thermal balance [17] is applied to a calorimeter designed to estimate the conductivity (Figure 13) of different materials following Fourier’s Law (Table 4) (To determine the heat transfer experiment, we based this study on Fourier’s law of thermal conduction, which considers a flat wall of thickness e and an average thermal conductivity λ.).

### 3.5. Analysis of Efflorescence in LTT

Efflorescence in LTT was analyzed to assess the presence and severity of salt deposits on the surface (Figure 14a). The analysis involved visual inspection and qualitative assessment of the efflorescence patterns, such as the type of salts present and their distribution.

Samples of LTT were collected and examined for efflorescence using established methods. The samples were observed under appropriate lighting conditions and evaluated for the presence of white or colored deposits on the surface. The intensity and extent of efflorescence were recorded and compared among different samples.

Additionally, chemical analysis techniques, such as ion chromatography or spectroscopy, may be employed to identify the specific salts responsible for the efflorescence.

The analysis of efflorescence in LTT provides valuable information on the potential for salt migration and deposition, which can have implications for the durability and aesthetics of the brick material.

One of the factors to consider in the production of LTT is the occurrence of efflorescence, which involves the migration of soluble alkalis and the subsequent dissolution of these alkalis with water, resulting in the formation of white carbonates. In this section, we will analyze the results of the analyses conducted on the samples after being exposed to the environment.

According to Lv et al. [6] in their study on “Efflorescence inhibition in sodium-based inorganic geopolymer coatings”, efflorescence is a spontaneous behavior in sodium-based geopolymers. The migration of soluble alkalis can lead to the dissolution of these alkalis in contact with water, resulting in the formation of white carbonates.

The results of the analyses will provide insights into the occurrence and severity of efflorescence in the LTT samples, which is an important factor to consider in assessing the durability and aesthetic quality of the material.

We proceed to perform an XRD analysis of the LTT prototype after a minimum time of 28 days has elapsed, during which it has been exposed to “normal” weathering conditions. The obtained results are shown in Table 5.

The first four crystalline phases (quartz, hematite, albite, and muscovite) correspond to minerals that should be present in the brick and are considered harmless.

The last two identified compounds (hydrated aluminum phosphate and hexahydrated magnesium and potassium sulfate) are hydrated phosphates/sulfates that may have formed due to moisture on the surface of the bricks.

It is important to note that efflorescence does not affect the mechanical strength of the bricks and can be eliminated by rainwater action without contaminating aquifers. Alternatively, it can be treated with hydrophobic siloxanes (Figure 14b), coated with a lime slurry or mortar, or even painted with silicate paints.

Once the thermal conductivity of the new material is obtained, the interior thermal comfort of a house in “Ciudad Victoria” (case study) is simulated using Ecodesigner software, an extension of the Archicad BIM program developed by Graphisoft. This process involved six phases: calibration of the location, climate, and environment; creation and definition of the physical characteristics of the materials; BIM modeling of the building; configuration of operation profiles for each area of the house; analysis and incorporation of infiltrations; simulation and results.

Referring to Section 3 of NEC-HS-EE (Ecuadorian Construction Standard–Habitability and Health–Energy Efficiency), the climatic housing zoning for the city of Loja falls under “climate zone 3”, referred to as “rainy continental” (From the climate file of the city of Loja, obtained from the weather station “Loja La Argelia LJ ECU”, it is determined that the Heating Degree Days (HDDs) have a value of 588, considering a base temperature for heating of 18 °C. On the other hand, the Cooling Degree Days (CDDs) have a value of 2339, with a base temperature of 10 °C, following the ASHRAE standard as a thermal criterion for climate classification. Analyzing the HDD 18 and CDD 10 data from the weather station and Table 1 of the NEC-HS-EE (page 12), it was determined that the city of Loja belongs to Climate Zone 3, within Zone 3C according to ASHRAE 90.1 standard, referred to as CONTINENTAL LLUVIOSA, meeting the thermal criterion of CDD 10 °C ≤ 2500 and HDD 18 °C ≤ 2000). Using the degree-day method (It shows, based on the geographical location, whether the building has heating or cooling requirements, or both. It is also defined as the temperature difference between the average outdoor air temperature over a 24 h period and a specified base temperature.) and the climatic data obtained from the website [18], which includes the annual averages from the LA ARGELIA LJ ECU, meteorological station the following values are provided in Table 6.

On the other hand, the “Climate Consultant 6” program is a tool used in this research as it allows for the decomposition of multiple climate variables into simple graphs. Both the “Climate Consultant 6” and “Ecodesigner” tools are compatible with the EPW extension; however, their functions are different. Through this complementary tool, we can more accurately analyze the climatic components within the EPW file for the climate of Loja. Additionally, this tool can be adapted to four different comfort models that align with the international ASHRAE standard. For the study of the rainy continental climate in Ecuador, the ASHRAE Standard 55 (It is considered in spaces with natural ventilation where occupants can open and close windows. Its thermal response depends in part on the external climate and may have a wider comfort range than in buildings with centralized HVAC systems. This model assumes that occupants adapt their clothing to the thermal conditions and are sedentary (1.0 to 1.3 MET). There should not be a mechanical cooling system in place, and this method is not applicable if a mechanical heating system is operating. Additionally, it considers a comfort range of 19.9 °C for lower comfort and 25.5 °C for maximum comfort according to the climate) was chosen, as the parameters to be analyzed in this research align with the suggestions provided by the standard.

The case study typology consists of a construction area of 40 m^2^ and is developed on a single-floor distribution as follows (Figure 15).

Dynamic thermal simulation is a valuable tool for predicting the long-term thermal performance of buildings (For the simulation, the Ecodesigner software from Archicad is used). This method utilizes detailed data in three key areas: building geometry and structure, climatic and location data, and simulation software algorithms (Table 7). Thanks to the precision of this data, results can closely reflect reality, enabling a better understanding of building behavior and optimization of time and resources. The obtained results are reliable enough to be used in future projects, and the methodology allows for a quick comparison of the BIM model’s capabilities in different scenarios, including variations in climate, location, and construction materials.

The construction of the different elements of the houses is carried out, including the floors, interior walls, and openings (The thermal conductivity values for each element were obtained from the Ecuadorian Construction Regulations (2018), while the infiltration levels were taken from the air tightness manual for buildings in Chile by Cite and Decon (2011)). The roof of the house was built with a 7 cm thick concrete slab.

This research evaluates potential errors in the structures and openings of the houses, considering aspects such as the materiality of window frames, the type of glass, and the level of protection established at 40% due to the use of fabric curtains. A specific schedule is set for window opening, from 2 PM to 3 PM, starting from 1 September to 31 July. High levels of infiltration, which are common in constructions in Ecuador, are considered. Subsequently, the houses are simulated, and the results are configured based on the dates that mark temperature peaks in each space. The data are used to determine if the houses comply with the comfort ranges established by the regulations applied to the city of Loja, which require a maximum temperature of 25 °C and a minimum of 18 °C, according to the comfort range calculation.

The requirements of the NEC-HS_EE are based on meeting specific thermal transmittance (U) values for the building envelope’s construction components. To verify compliance with these regulations, it is essential to identify the habitable and non-habitable zones of the house and analyze the construction systems that make up the envelope, along with their components and thermal properties. The objective of this analysis is to determine if the houses meet the parameters established by the regulations.

Next, we detail the components of each construction package of the envelope, along with their descriptive characteristics.

The construction elements of the house, such as walls and floors, comply with the thermal transmittance (U) requirements established by the NEC regulations (Table 8). However, the roofs do not meet the standards set by the regulations.

Furthermore, the house has windows with natural aluminum frames and clear 4 mm glass on the front and back facades. According to the NEC-HS_EE regulations, the glazed areas should not exceed 40% of the net area of the facade. The calculation of the glazed area is shown in the following table (Table 9).

The analysis of the hygrothermal comfort of the houses reveals that the appropriate temperatures for thermal comfort are not met, causing discomfort to the inhabitants due to constant temperature fluctuations. These fluctuations are caused by using materials with high thermal transmittance in the building envelope and interior spaces (Table 10). The green box in Table 10 highlights the minimum thermal requirements (U and R) for non-air-conditioned elements in climate zone 3, emphasizing the performance criteria that must be met to improve indoor thermal comfort.

The building envelope plays a crucial role in regulating internal and external temperatures. In winter, internal heat dissipates through the envelope, while in summer, external heat penetrates and accumulates in the house.

Regarding the floor, Type I houses have issues with their construction packages, particularly in areas with ceramic flooring, which creates a cool environment that is not suitable for houses in Ciudad Victoria (Loja). Instead, materials such as wood, parquet, or laminate flooring are needed to maintain a warm environment.

The compliance analysis of the NEC-HS-EE regulations demonstrates that the house does not meet the specified values for its respective climatic zone (climate zone 3), resulting in issues with habitability and comfort. Problems associated with the materials used in the interior spaces and building envelope have been identified. Therefore, it is suggested to implement strategies focused on improving the construction materials of the house to enhance the quality of the interior spaces (Table 11).

### 3.6. Results of the Application

To improve the habitability of the house, it is proposed that the materiality of the construction packages be modified based on residential design guidelines for the city of Loja.

For the roof, it is suggested that a 5 mm asphalt membrane be adhered to the concrete slab for insulation and waterproofing. Additionally, a 10 cm mineral wool sheet should be added to the slab to control excessive heat loss and gain, and a 9 mm plasterboard should be proposed.

For the walls, the use of a 2–4 cm geopolymerized brick veneer is recommended, selected for its ability to maintain the interior temperature in relation to the exterior temperature and conditioning of the houses.

Regarding the floors, due to low temperatures in the bedrooms, a floating floor is suggested. This material can store solar gain during winter days, providing warmth, and stay cool during summer nights, regulating the temperatures of the bedrooms (Table 12).

On the other hand, to determine the actual contributions using only the LTT in the envelope and, as a possible extension of this research to the use of this material in floors, the simulation is performed without considering improvements in the roof of the Type I house, and considering the change in material thickness, as depicted in the following figures (Figure 16 and Figure 17).

## 4. Conclusions

The XRD and XRF analyses confirmed that the ceramic waste meets the necessary conditions for geopolymerization, with sufficient quartz (SiO_2_) and alumina (Al_2_O_3_) levels to support high-strength aluminosilicate formation. The geopolymerization process using brick waste was found to be viable, producing materials with satisfactory mechanical and thermal properties. The optimal mix was 12.5 M–26%CS–150 °C, achieving a peak optimization factor of 14.84 MPa/kg. The produced LTT bricks meet the compressive strength requirement of 6 MPa set by NTE INEN 297, reaching an average of 6.93 MPa, although further improvements are needed to meet flexural strength standards. Heat transfer tests and energy simulations suggest efficient thermal behavior, making LTT suitable for architectural envelopes. Efflorescence does not compromise mechanical integrity and can be addressed with hydrophobic treatments.

## Figures and Tables

**Figure 1 materials-18-00103-f001:**
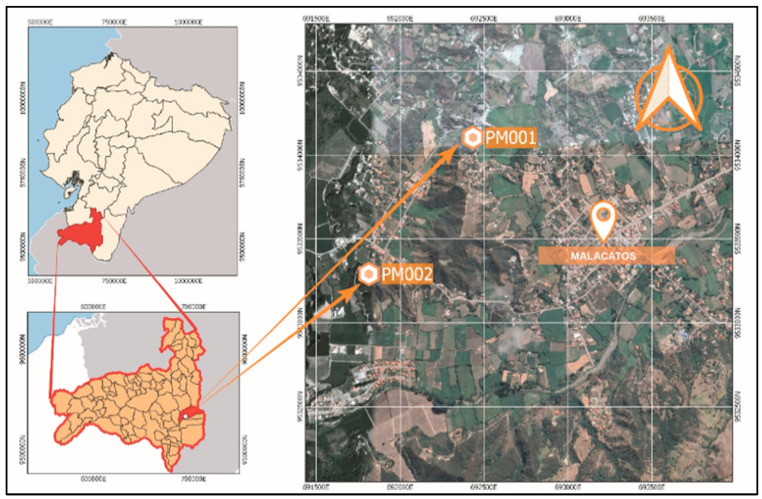
Location map of producers PM001 and PM002 in the city of Malacatos, Loja Province, Ecuador.

**Figure 2 materials-18-00103-f002:**
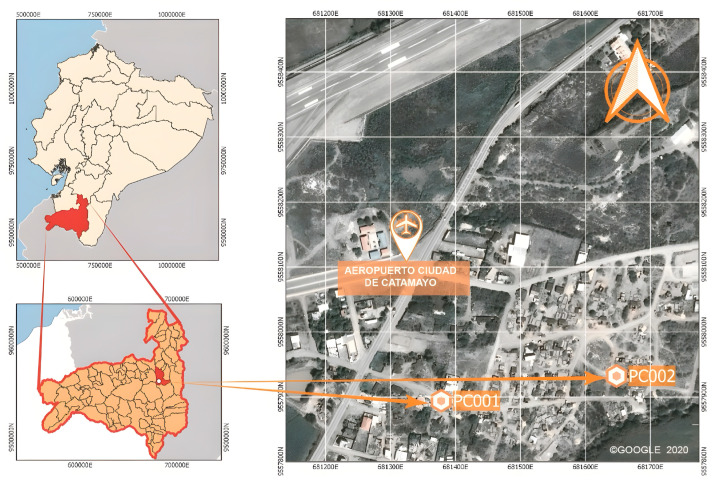
Location map of producers PC001 and PC002 in the city of Catamayo, Loja Province, Ecuador.

**Figure 3 materials-18-00103-f003:**
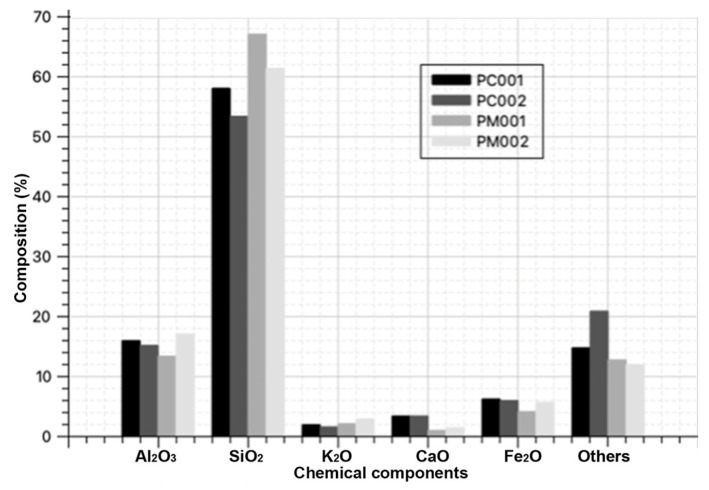
X-ray fluorescence percentage by factory code.

**Figure 4 materials-18-00103-f004:**
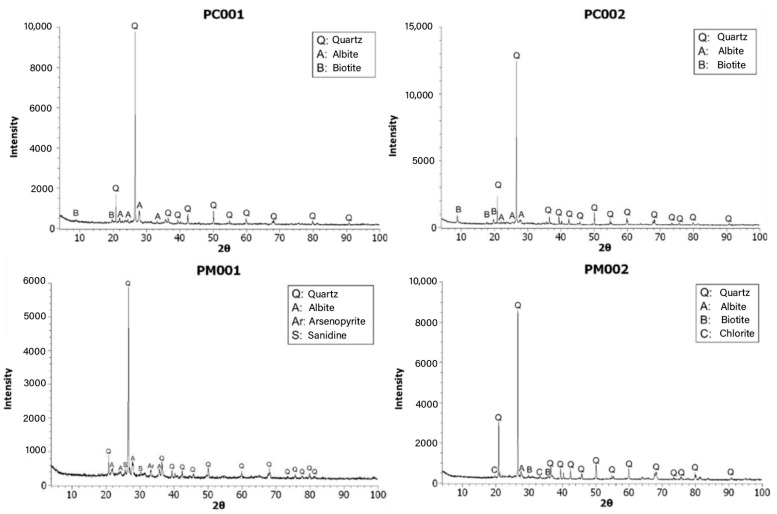
X-ray diffraction: PC001, PC002, PM001, PM002.

**Figure 5 materials-18-00103-f005:**
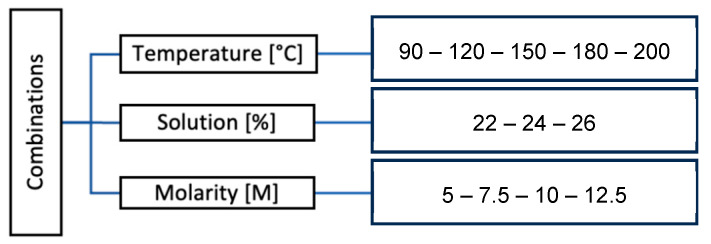
Determining optimal mixture variables for geopolymerization.

**Figure 6 materials-18-00103-f006:**
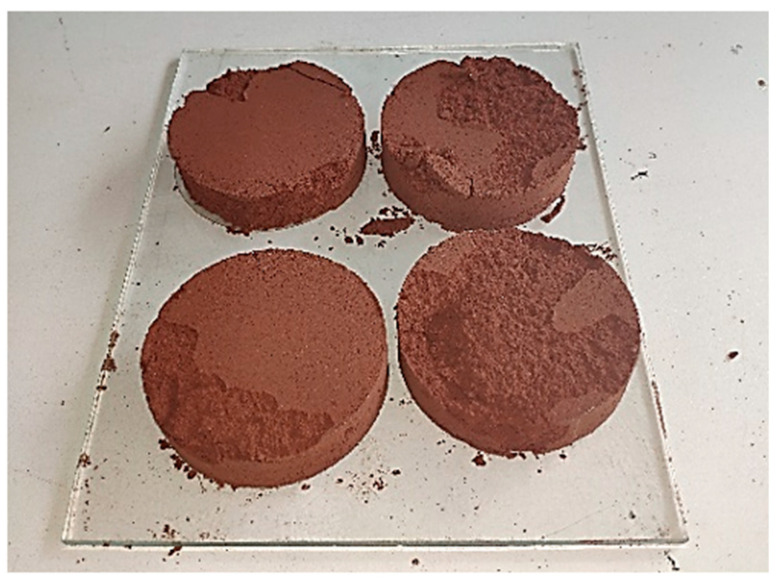
Discs with pressures lower than 39.2266 N.

**Figure 7 materials-18-00103-f007:**
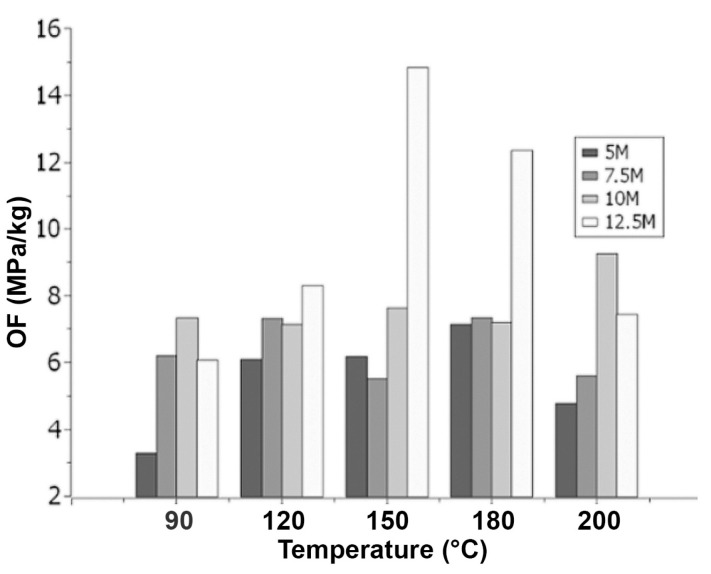
Scheme 26 % SC.

**Figure 8 materials-18-00103-f008:**
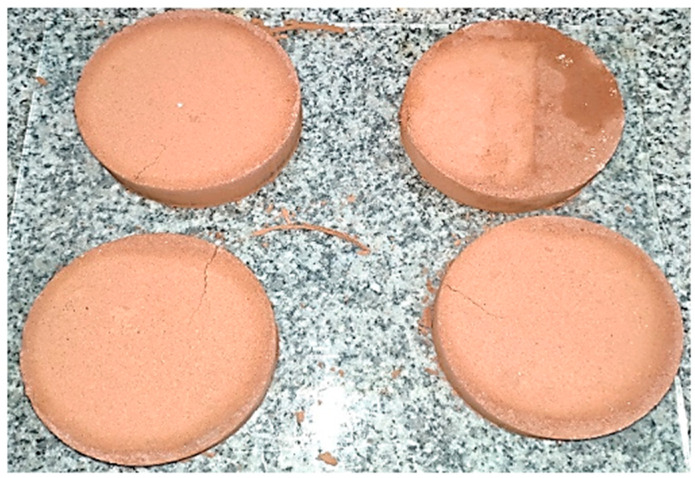
Cracks generated due to excessive curing temperature.

**Figure 9 materials-18-00103-f009:**
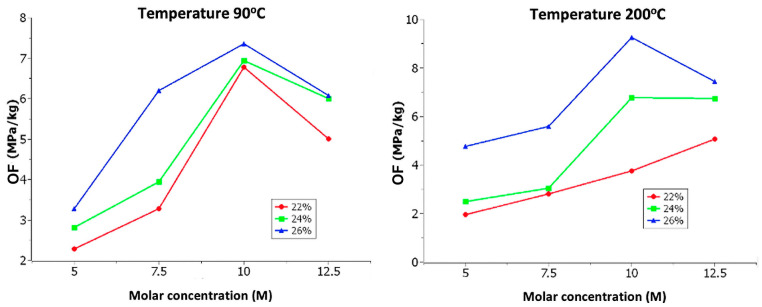
Effect of molar concentration: 90 °C and 200 °C.

**Figure 10 materials-18-00103-f010:**
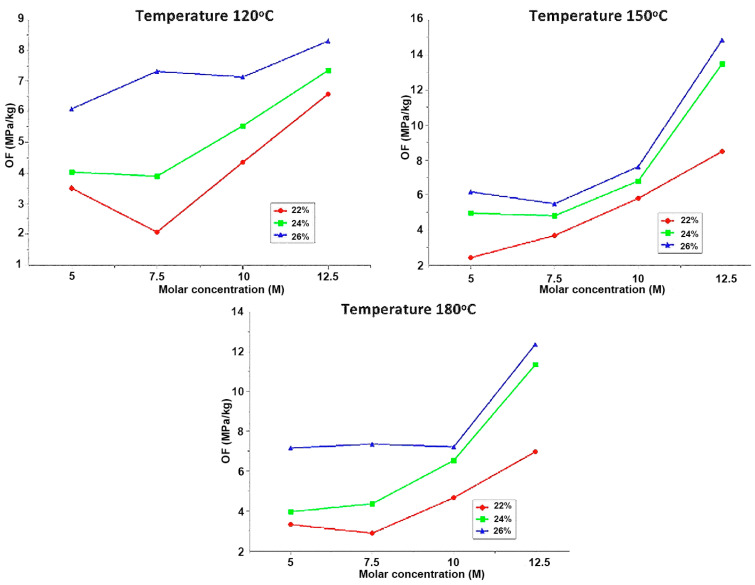
Effect of molar concentration at: 120 °C, 150 °C and 180 °C.

**Figure 11 materials-18-00103-f011:**
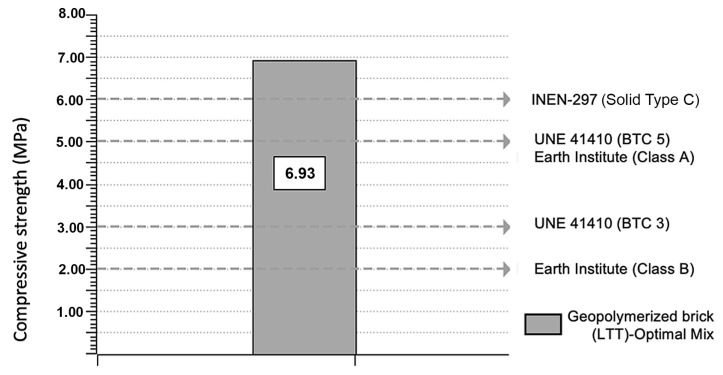
Compressive strength results.

**Figure 12 materials-18-00103-f012:**
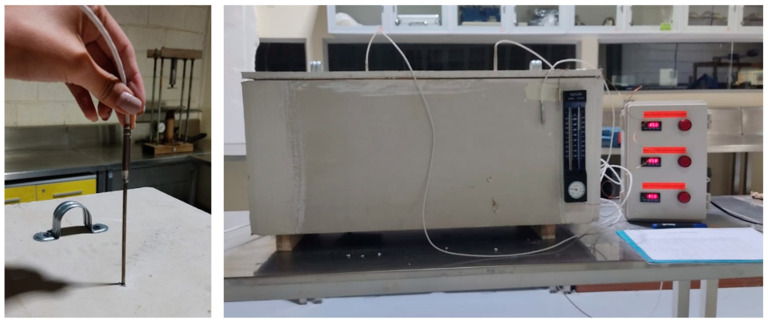
Data collection with thermocouple at intervals of 1, 30, and 60 min.

**Figure 13 materials-18-00103-f013:**
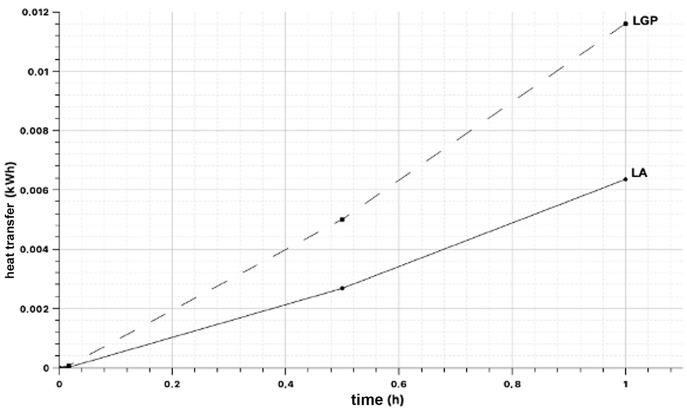
Heat transfer graph.

**Figure 14 materials-18-00103-f014:**
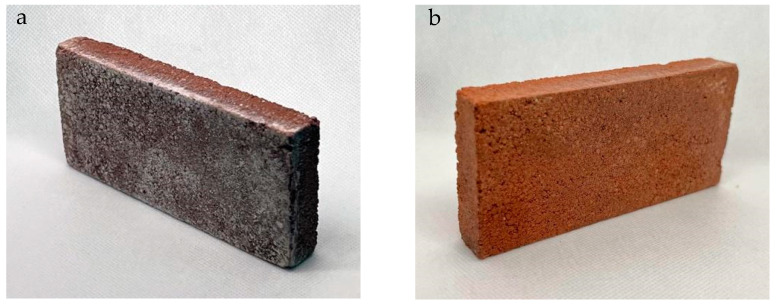
(**a**) LTT prototype with efflorescence. (**b**) LTT with silicate paint protection.

**Figure 15 materials-18-00103-f015:**
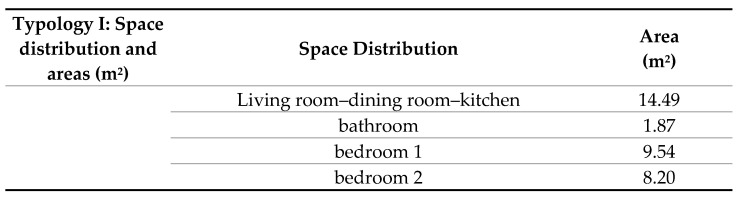
Single story—elevation.

**Figure 16 materials-18-00103-f016:**
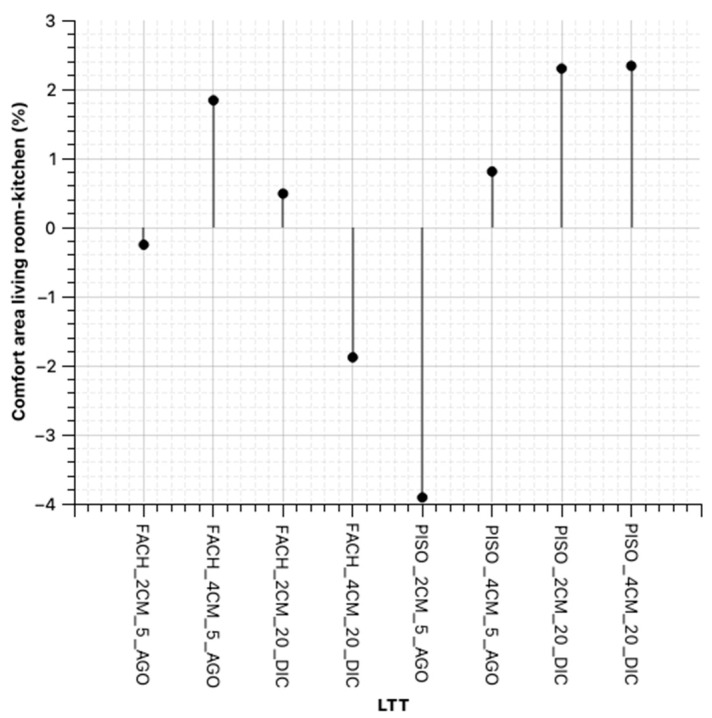
Scheme 19. Space: living room—kitchen (98–24.98 °C) applying LTT, compared to the base case.

**Figure 17 materials-18-00103-f017:**
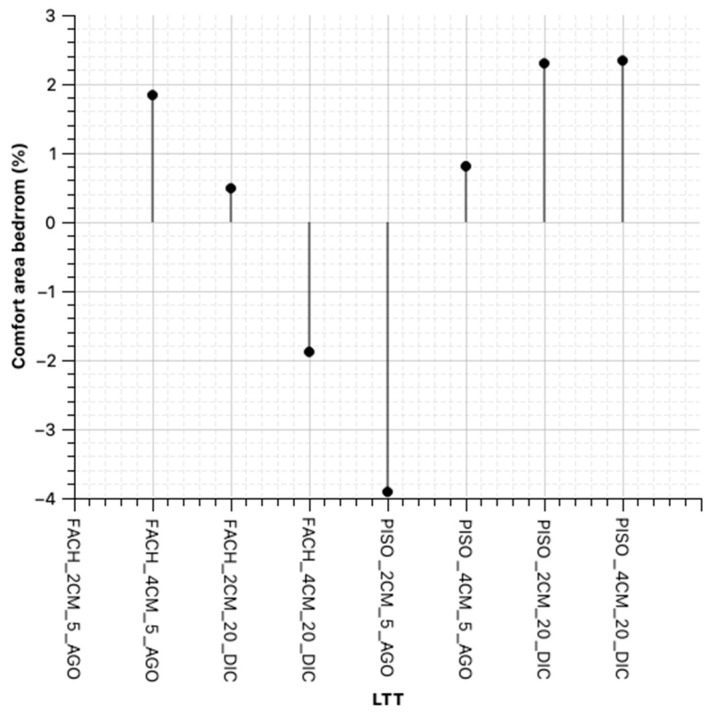
Scheme 19. Space: Bedroom (98–24.98 °C) applying LTT, compared to the base case.

**Table 1 materials-18-00103-t001:** X-ray fluorescence results.

Sample	AL_2_O_3_ %	SiO_2_ %	K_2_O %	CaO %	Fe_2_O_3_ %	Others %	Total %	Oxides > 70%
PC001	15.90	58.00	1.86	3.34	6.18	14.72	100.0	80.08
PC002	15.10	53.30	1.53	3.35	5.90	20.82	100.0	74.30
PM001	13.30	67.00	2.04	0.95	4.04	12.67	100.0	84.34
PM002	17.00	61.30	2.84	1.39	5.59	11.88	100.0	83.89

**Table 2 materials-18-00103-t002:** Molarity dosing.

Capacity of Beaker	Molarity	Amount of NaOH in 1000 (mL)	Amount of NaOH in 100 (mL)
A(ml)	B(M)	C (g)	D (g)
250	12.5	500	125
	10	400	100
	7.5	300	75
	5	200	50
100	12.5	500	50
	10	400	40
	7.5	300	30
	5	200	20

**Table 3 materials-18-00103-t003:** Solution content in the soil–solution combination.

Solution Content	Soil Content	Soil-Solution Content
(%)	(%)	(%)
22	78	100
24	76	100
26	74	100

**Table 4 materials-18-00103-t004:** Heat transfer test results.

Sample Code	SE-T1 (°C)	SR-T2 (°C)	TE (°C)	Time t (h)	Conductivity (W/m·K)	Material Area A (m^2^)	Material Thickness e (m)	Heat Transfer Coefficient Q’ (kW)	Heat Transfer Q (kWh)
LGP-1	16.70	15.60	16.40	0.017	0.954	0.090	0.020	0.0047	0.000052
LGP-1	19.00	15.50	15.60	0.500	0.954	0.090	0.020	0.0098	0.004883
LGP-1	19.80	15.60	15.50	1.000	0.954	0.090	0.020	0.0117	0.011718
LGP-2	17.40	15.70	15.80	0.017	0.954	0.092	0.021	0.0046	0.000077
LGP-2	19.40	15.60	15.80	0.500	0.954	0.092	0.021	0.0102	0.005088
LGP-2	19.90	15.60	15.70	1.000	0.954	0.092	0.021	0.0115	0.011515
LA-1	19.80	19.40	19.90	0.017	0.800	0.058	0.043	0.0004	0.000007
LA-1	25.10	19.00	19.40	0.500	0.800	0.058	0.043	0.0066	0.003291
LA-1	26.10	19.00	19.00	1.000	0.800	0.058	0.043	0.0077	0.007661
LA-2	20.40	19.40	19.70	0.017	0.800	0.058	0.042	0.0011	0.000019
LA-2	23.00	19.30	19.40	0.500	0.800	0.058	0.042	0.0041	0.002044
LA-2	24.00	19.50	19.30	1.000	0.800	0.058	0.042	0.0050	0.004971

**Table 5 materials-18-00103-t005:** Mineral phases (chemical compounds) detected in the sample.

Mineral Compound	Chemical Formula	Semi-Quantification (%)
quartz	SiO_2_	73
hematite	Fe_2_O_3_	1
albite	NaAlSi_3_O_8_	1
muscovite	KAl_2_(Si_3_Al)O_10_(OH,F)_2_	20
hydrated aluminum phosphate	Al_5_[(PO_4_)_2_[(P,S)O_3_(OH,O)]_2_F_2_(OH)_2_(H_2_O)8•6.48(H_2_O)	1
hexahydrated magnesium potassium sulfate	K_2_Mg (SO_4_)_2_•6(H_2_O)	4

**Table 6 materials-18-00103-t006:** Summary table of the climate in the city of Loja [18].

Ecuador Climate Zone	January	February	March	April	May	June	July	August	September	October	November	December	Unit
Global horizontal radiation *	372	347	414	407	368	356	302	380	420	404	458	400	Wh/m^2^
Direct normal radiation *	212	154	249	275	270	312	204	309	298	226	319	266	Wh/m^2^
Diffuse radiation *	224	237	240	215	201	160	174	172	224	242	221	215	Wh/m^2^
Global horizontal radiation **	1049	1075	1077	1045	988	930	963	995	1055	1071	1091	1039	Wh/m^2^
Direct normal radiation **	1001	998	1011	1008	984	983	994	988	991	985	1053	1019	Wh/m^2^
Diffuse radiation **	538	549	558	532	490	450	458	494	546	567	543	522	Wh/m^2^
Global horizontal radiation ***	4552	4214	4981	4850	4358	4197	3567	4511	5031	4890	5582	4898	Wh/m^2^
Direct normal radiation ***	2592	1872	2995	3275	3194	3683	2412	3668	3577	2736	3899	3262	Wh/m^2^
Diffuse radiation ***	2745	2881	2895	2563	2385	1885	2053	2051	2691	2927	2697	2638	Wh/m^2^
Horizontal global illumination *	41,274	38,102	45,280	44,837	41,338	39,798	33,943	41,940	45,970	44,115	50,230	44,414	lux
Direct normal illumination *	20,143	14,321	23,495	25,998	26,437	31,269	20,209	30,357	28,057	21,064	29,851	25,404	lux
Dry bulb temperature ****	14	15	15	14	15	14	14	14	15	15	14	15	°C
Dew point temperature ****	11	12	12	11	12	11	10	9	12	11	10	12	°C
Relative humidity ****	84	86	83	85	85	81	79	70	80	77	81	82	%
Wind direction *****	270	270	90	90	90	90	90	90	90	90	270	270	degrees
Wind speed ****	1	1	2	1	1	2	3	2	2	1	0	1	m/s
Floor temperature *,***	14	14	14	15	15	15	15	15	15	14	14	14	°C

*—Average per hour; **—Maximum per hour; ***—Daily total average; ****—Monthly average; *****—Monthly mode.

**Table 7 materials-18-00103-t007:** Construction packages. Infiltration level (The construction packages used in both housing typologies are summarized in the table, considering the thermal conductivity values taken from the Ecuadorian construction standard NEC (2018) and the infiltration levels from the air tightness manual for buildings in Chile [19].

Construction Package		Components	Thickness (cm)	Conductivity (W/°K)	Infiltration Level (L/sm^2^)
roofing	reinforced concrete slab	reinforced concrete	7	2.3	4.40
walls	concrete block	exterior plastering	1	0.5	4.40
		concrete block	10	0.62	
		interior plastering	1	0.72	
floor	wood	hardwood	0.7	0.18	1.10
	ceramic	ceramic	0.5	1.75	1.10
doors	metal	steel	0.03	50	4
		air (R 0.15 m^2^k/w)	0.1		
		steel	0.03		
	wood	painted oak	4.2	0.19	
windows	single-pane glass	clear glass	0.3	0.9	4

**Table 8 materials-18-00103-t008:** Construction packages and thermal properties.

	Construction Package	Components	Thickness (cm)	Conductivity (W/m^2^K)	Current U-Value (W/m^2^K)	Reference U-Value (W/m^2^K)	Compliance with NEC
unconditioned living space ceilings	e = 8.2 cm	reinforced concrete	7.0 cm	2.3	4.7	2.35	Does not comply
		plastering	1.2 cm	0.81			
Walls of habitable spaces—non-conditioned—above ground level	e = 12.0 cm	plastering	1.0 cm	0.116	2.89	2.90	Complies
		concrete block	10 cm	0.62			
		plastering	1.0 cm	0.116			
Floors of habitable spaces—non-conditioned	e = 30 cm	ceramic	0.7 cm	1.75	2.82	3.2	Complies
		subfloor	8 cm	2.50			

**Table 9 materials-18-00103-t009:** Percentage of insulated surface—net wall area of a Type I residential unit.

Elevation	Total Facade Area (m^2^)	Area of Glazed Surface (m^2^)	Percentage of Glazed Surface (%)	Compliance with NEC–HS-EE
Front elevation	15.12	3.29	27.1%	Complies
Rear elevation	12.12	3.58	23.6%	Complies

**Table 10 materials-18-00103-t010:** Envelope requirements for climate zone 3.

Opaque Elements	Air-Conditioned		Not Air-Conditioned		Not Habitable	
	Maximum Assembly	Minimum Insulation Resistance Value	Maximum Assembly	Minimum Insulation Resistance Value	Maximum Assembly	Minimum Insulation Resistance Value
Roofs	U-0.273	R-3.5	U-2.9	R-0.89	U-4.7	R-0.21
Walls above ground level	U-0.592	R-1.7	U-2.35	R-0.36	U-5.46	NA
Walls below ground level	C-6.473	NA	C-6.473	NA	C-6.473	NA
Floors	U-0.496	R-1.5	U-3.2	R-0.31	U-3.4	NA
Opaque doors	U-2.839	NA	U-2.6			
Windows	Maximum transmittance	Maximum SHGC assembly	Maximum transmittance	Maximum SHGC assembly	Maximum transmittance	Maximum SHGC assembly
Vertical glazing ≥ 45 degrees	U-3.69	SHGC-0.25	U-5.78	SHGC-0.82	U-6.81	NA
Horizontal glazing < 45 degrees	U-6.64	SHGC-0.36	U-6.64	SHGC-0.36	U-11.24	NA

**Table 11 materials-18-00103-t011:** Compliance values for building envelope according to NEC-HS-EE.

Building Envelope—Type I Residential Unit				
Typology	Component	Current U-Value(W/m^2^K)	Required U-Value (W/m^2^K)	Compliance
roofing	concrete slab 7 cm thick	4.7	2.35	Does not comply
walls	concrete block	2.89	2.90	Complies
Intermediate floor	ceramic	2.82	3.20	Complies

**Table 12 materials-18-00103-t012:** NEC compliance verification: proposed Typology I.

Building Envelope, Typology I Residence				
Typology	Component	Current U Value (W/m^2^k)	Required Value (W/m^2^k)	Compliance
roofing	concrete slab 7 cm thick	0.643	U-2.35	Complies
walls	concrete block	2.89	U-2.90	Complies
floor	ceramic	2.31	U-3.20	Complies

## Data Availability

The data generated during this study are not publicly available now, as the research is ongoing and improvements to the formulations are still under development. However, specific data or additional information can be provided to the corresponding author upon reasonable request.

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
