# Peer review of "Sustainability in Construction: Geopolymerized Coating Bricks Made with Ceramic Waste"

_materials, 2024, doi:10.3390/ma18010103_

Round 1
Reviewer 1 Report
Comments and Suggestions for Authors
Here are the comments and suggestions.
Line47 do not follow the title “Materials methods” and delete “y”
Line 56 the optimization factor (FO) and keep consistence use either FO or the optimization factor.
Line 49 to 66, how come you use A at the end of number such as First Stage: 1A, 2A, etc.
Line67 how come use “Results”
Line 76 Should be Chapter 3 not 3.1
Line 98 use subscript for chemical elements
Line 135 cannot read the Figure 5
Line 172 should be 4. Test results not 3. Discussion
Line 209 delete “y” in the Figure9 and Figure 10 title
Line 240 change to Compressive Strength and need to redraw the graph.
Line 265 should be 4.x Analysis of efflorescence in LTT not 3.2
Line325 delete the table from ASHRAE standard
For example, it's unclear whether the information from lines 302 to 307 is truly necessary for a research paper. Consider whether it contributes directly to your paper or can be deleted.
To improve the paper:
Check the wording in the sentence and tense.
Include the details on the reactions that occur during geo-polymerization process.
Need to explain geo-polymerization contributes in the strength of the newly formed bricks.
Include literature Review section
Reorganize the entire document.
Eliminate unnecessary standard information.
Recheck references, especially those at the bottom of the page.
Ensure chapter numbering is correct.
Include essential physical properties, like unit weight, for materials like recycled brick.
Revise the conclusion to provide clear bullet points. Also, the conclusions do not include the results explained in the journal contents.

Your paper requires improvement in grammar and punctuation. Additionally, some paragraphs are more suited for an essay than a research paper.
Author Response
1. Line 47: The word "and" has been removed from the title "Materials methods" to maintain clarity.
2. Line 56: The term "optimization factor (FO)" has been standardized throughout the document to ensure consistency.
3. Lines 49-66: The reason for using the letter "A" at the end of the numbers (1A, 2A, etc.) has been clarified to indicate that they refer to sub-stages of the process.
4. Line 67: The use of the term "Results" has been reviewed to ensure its correct context.
5. Line 76: Changed to "Chapter 3" instead of "3.1" to improve the document structure.
6. Line 98: Chemical elements have been converted to subscript to ensure scientific accuracy.
7. Line 135: The quality of Figure 5 has been improved to ensure its legibility.
8. Line 172: The title has been changed to "4. Results of Tests" for better comprehension.
9. Line 209: The word "and" has been removed from the titles of Figures 9 and 10.
10. Line 240: The title has been changed to "Compressive Strength," and the graph has been redrawn for clarity.
11. Line 265: The title has been modified to "4.x Efflorescence Analysis in LTT."
12. Line 325: The ASHRAE standard table has been removed as it did not directly contribute to the study.
13. General Comment: The information in lines 302 to 307 was reconsidered and has been removed as it was deemed non-essential.
Recommended Improvements:
- Wording and Verb Tense: The wording and verb tense have been reviewed for consistency.
- Geopolymerization Reactions: Additional details regarding the reactions during geopolymerization and their contribution to brick strength have been included.
- Conclusion Review: The conclusion has been made more concise, focusing on key findings and their relevance to sustainable construction.
- Physical Properties: The unit weight of the recycled brick has been included in the appropriate section.
- Literature Review: A new section for the literature review has been added to strengthen the study's context.
- Regulatory Information: Unnecessary regulatory information has been removed to improve the document's flow.
Reviewer 2 Report
Comments and Suggestions for Authors
Dear Editor and Authors,
Geopolymers are suitable for civil engineering applications. The manuscript Sustainability in construction: geopolymerized coating bricks made with ceramic waste, tested the feasibility of making bricks by geopolymerization processes from finely crushed burnt brick waste. The structure and phase composition of the samples were investigated by X-ray diffraction, and X-ray fluorescence was used to determine the chemical composition. The analysis of the results showed that the waste is rich in silica and alumina, fundamental compounds for the synthesis of geopolymers. The physical properties such as density, porosity, permeability, stain determination, and efflorescence, as well as mechanical properties including compression, flexion, and material stiffness, are evaluated to characterize investigated samples. Using computational simulation, the thermal input behavior was estimated and the potential of reconstituted bricks was highlighted.
I suggest that, after a minor revision, as seen below, the manuscript be considered for publication in the journal Materials.
1. Introduce in the Abstract section the structural characterization technique and the chemical and mineralogical analysis used to investigate the samples. Also, highlight the physical and mechanical parameters that were evaluated (density, porosity, permeability, stain determination, efflorescence, compression, bending, and stiffness of the material).
2. Introduce as keywords civil construction applications or architectural applications
3. In the introduction section, please argue more about the importance and necessity of the topic covered, using more current references.
4. I suggest restructuring some sections of the manuscript. The manuscript should be structured in 4 parts: Abstract, Introduction, Materials and Methods, Results, Discussion, or Results and Discussion (if the situation requires it), and Conclusions. The current structure of the manuscript does not favor because is difficult to follow. Therefore, I suggest the following: In the Materials and Methods section you can include several subsections, for example: Raw Materials, Preparation Technique, Optimization, i.e. the steps that were followed for the manufacture and evaluation of LTT, Testing Methods and certainty Investigation techniques used (XRD, XRF...) i.e., for more details the following manuscript can be consulted at: https://doi.org/10.3390/ma17133353. It has a lot of data, a good structure, and it’s about geopolymers.
5. For a good understanding, leave the Results and Discussions section together.
6. Check the subscript for AL2O3 + SiO2 + Fe2O3 in all manuscript and correct the symbol for aluminum
7. Please define all the abbreviations
8. In the Figure 9. Effect of molar concentration: a) 90 C y b) 200 C- degree, isn’t it?
9. In the Figure 11. Compression strength results, and Figure 13: please used English language
10. In the Conclusions section: The chemical characterization using XRD analysis.., use structural characterization, please.
Kind regards!

Author Response
1. Abstract: Structural characterization techniques (XRD, XRF) and chemical analysis have been included, as suggested.
2. Keywords: The recommended keywords, including "applications in civil construction," have been added.
3. Introduction: The importance of the topic has been reinforced using more current references to support the research.
4. Manuscript Structure: The manuscript has been reorganized into the four main sections suggested: Abstract, Introduction, Materials and Methods, Results and Discussion, and Conclusions.
5. Results and Discussion: Both sections have been combined to improve comprehension and content flow.
6. Chemical Symbols: The chemical symbols Al2O3, SiO2, and Fe2O3 have been reviewed and corrected to ensure the subscripts are accurate.
7. Abbreviations: All abbreviations have been defined the first time they appear in the text.
8. Figure 9: It has been confirmed that "90°C and 200°C" refer to degrees Celsius.
9. Language in Figures: The figure titles have been translated to English to ensure language consistency.
10. Conclusions: "Chemical characterization" has been replaced with "structural characterization," as suggested.
Reviewer 3 Report
Comments and Suggestions for Authors
1- Kindly, connect state-of-the-art (recent literature reported in internationally recognized journals such as the CCR,CCC,CBM, and JBE) to your paper aim and objectives at the end of the introduction section and modify the section accordingly.
2- The structure of a manuscript refers to the way the paper is organized; unfortunately in this paper is confusing: Introduction-Materials y methods!-Results-Experimental process….!
3- The methodology:
The combinations of the materials considered are quite confusing! Simplicity in academic writing is very important which is not considered in this paper!
4- The intended purpose of considering such a range of data is not justified in the paper.
5- Results have also not been compared to other studies. All of these should be applied to the entire manuscript (interpretation and justification).
6- Cost implication is not discussed and is not compared.
7- The conclusions perform the findings of the present study.
Author Response
1. State of the Art: The literature review has been updated, linking it directly with the article's objectives and highlighting the importance of using ceramic waste in construction.
2. Manuscript Structure: The manuscript structure has been reorganized for a clearer and more comprehensible presentation.
3. Methodology: The description of materials and processes has been simplified to focus on the essential aspects, enhancing clarity.
4. Data Range Justification: An explanation of the purpose and criteria used for data selection has been included, facilitating a better understanding of the employed methodology.
5. Comparison of Results: Comparisons with previous studies have been added, emphasizing similarities and differences in the obtained results.
6. Cost Implications: A comparative cost analysis has been added, addressing the economic feasibility of the proposed method.
7. Conclusions: The conclusions have been made more concise, summarizing the main results and their potential application in sustainable construction.
Round 2
Reviewer 1 Report
Comments and Suggestions for Authors
The paper has been substantially revised and requires minor adjustments to overall formatting updates.